# Obesity-Associated ECM Remodeling in Cancer Progression

**DOI:** 10.3390/cancers14225684

**Published:** 2022-11-19

**Authors:** Junyan Li, Ren Xu

**Affiliations:** 1Markey Cancer Center, University of Kentucky, Lexington, KY 40506, USA; 2Department of Pharmacology and Nutritional Sciences, University of Kentucky, Lexington, KY 40506, USA

**Keywords:** ECM remodeling, fibrosis, adipocyte plasticity, obesity, breast cancer

## Abstract

**Simple Summary:**

Accumulated evidence has demonstrated that adipocytes can transform or de-differentiate into myofibroblast/fibroblast-like cells, which play vital roles in obesity-related extracellular matrix (ECM) remodeling and cancer progression. This review summarizes recent progress in adipocyte plasticity during obesity-related cancer progression and the function and regulation of obesity-associated ECM remodeling in cancer progression.

**Abstract:**

Adipose tissue, an energy storage and endocrine organ, is emerging as an essential player for ECM remodeling. Fibrosis is one of the hallmarks of obese adipose tissue, featuring excessive ECM deposition and enhanced collagen alignment. A variety of ECM components and ECM-related enzymes are produced by adipocytes and myofibroblasts in obese adipose tissue. Data from lineage-tracing models and a single-cell analysis indicate that adipocytes can transform or de-differentiate into myofibroblast/fibroblast-like cells. This de-differentiation process has been observed under normal tissue development and pathological conditions such as cutaneous fibrosis, wound healing, and cancer development. Accumulated evidence has demonstrated that adipocyte de-differentiation and myofibroblasts/fibroblasts play crucial roles in obesity-associated ECM remodeling and cancer progression. In this review, we summarize the recent progress in obesity-related ECM remodeling, the mechanism underlying adipocyte de-differentiation, and the function of obesity-associated ECM remodeling in cancer progression.

## 1. Introduction

Obesity is epidemiologically linked to the development of 13 different types of cancer, including breast cancer, gastric cancer, pancreas cancer, and melanomas [1]. It is estimated that obesity is associated with 20% of cancer-related deaths [2]. Obesity-dependent cancer development and progression involve inflammation, adipokines, microbiota, hyperinsulinemia, and IGF-1 signaling, which has been extensively summarized and reviewed [3,4,5]. In this review, we focused on recent progress in the function and regulation of obesity-associated extracellular matrix (ECM) remodeling and adipocyte plasticity during cancer progression.

The development of obesity requires continuous and adequate energy availability. Adipose tissue, as an energy storage and endocrine organ, is emerging as an essential player for systemic metabolic homeostasis. A key factor in the development from lean to obese is the dynamic response of adipose tissue to excessive nutrients. The direct and indirect interaction between adipose tissue and tumors is crucial for obesity-associated cancer development and progression [6]. It is well-established that adipose tissue is an active endocrine organ; adipokines and chemokines secreted by adipocytes serve as endocrine and paracrine cues to induce cancer progression. Adipose tissue also enhances free fatty acid uptake in cancer cells, facilitates the immune escape, and contributes to the fibrosis process in the tumor microenvironment [7,8].

Obese adipose tissue features an excessive deposition of the ECM components caused by an imbalanced synthesis of the fibrous components (such as collagen I, III, and VI) and matrix metalloproteinases (MMPs) [7]. Adipocytes, myofibroblasts, and fibroblasts are the major cellular components in the adipose tissue responsible for ECM production. Excessive collagen deposition and enhanced collagen alignment in adipose tissue have been observed in various models of obesity [9,10,11,12,13]. The connection between the fibrotic pathology and obesity-related metabolic disorders has been established [14,15]. Tissue fibrosis is a trigger for solid tumor development and progression [16]. Tumor migration/invasion and poor patient survival are related to tissue fibrosis and interstitial stiffness [17]. Recent studies have demonstrated that obesity-associated ECM deposition and alignment promote cancer development and progression [14,15].

Adipocyte differentiation was considered to be a terminal differentiation. However, recent studies have shown that adipocytes maintain plasticity and can de-differentiate into pre-adipocytes or fibroblast-like cells [10,18,19,20]. Adipocyte-derived fibroblasts were first reported upon the co-culture of 3T3-F442A adipocytes with breast cancer cells in vitro [21]. The adipocytes acquired a fibroblast-like morphology featuring an enhanced secretion of fibronectin and collagen I as well as an increased expression of the fibroblast-like biomarker FSP1 after co-culturing [21,22]. Data from lineage-tracing mouse models and a single-cell analysis showed that the de-differentiation of adipocytes occurs during mammary gland development, wound healing, and cancer development [13,21,23,24,25,26]. Notably, adipocytes adopted a fibroblast-like phenotype in obese mice with an enhanced expression of ECM-associated genes and ECM remodeling enzymes [10], suggesting that adipocyte plasticity is involved in obesity-associated fibrosis in adipose tissue.

The ECM is a major component in the tumor microenvironment that controls cancer development and progression [27,28,29]. Obesity-associated tumor development is partially driven by ECM remodeling in adipose tissue, which involves an increased deposition of the ECM and the enhanced crosslinking of collagen fibers [14,15]. Collagen VI has been identified from obese mammary fat tissue as the driver of cancer cell migration and invasion [14]. Evidence from tissue culture experiments suggests that an obese-associated mechanic alteration of the ECM in adipose tissue increased the malignant potential of mammary epithelial cells [15]. These studies demonstrate that the biochemical and biophysical cues from an altered ECM in adipose tissue promote tumorigenesis and enhance cancer cell invasion and migration [15].

## 2. Obesity-Associated Fibrosis and ECM Remodeling

Many ECM molecules and ECM-related enzymes have been identified in adipose tissue, including collagen [11,30,31,32], MMPs [33], and fibronectin [34,35]. Given the important function of the ECM in regulating cellular functions and differentiation, it is important to identify the components that are involved in obesity-associated ECM reorganization [7]. Among those ECM components, collagen and fibronectin are the most abundant ECM proteins in adipose tissue [36]. MMPs cleave collagenous proteins to enable an ECM turnover; MMP expression is also dysregulated in adipose tissue during obesity development [7] (Table 1). The deregulation of these three components disrupts ECM hemostasis in adipose tissue, subsequently enhancing ECM deposition and fibrosis [7,8,22].

### 2.1. Collagen

Collagen is the major structural ECM component in adipose tissue. The increased expression and deposition of collagen I and collagen VI feature in obese and cancer-associated ECM remodeling [14,15,37,38,39]. Collagen I inhibits adipogenic differentiation via YAP activation whilst promoting myofibroblast differentiation in adipose tissue [37,38]. Collagen VI is highly expressed in adipocytes and its expression is significantly induced in adipose tissue during obese development [11]. Using a Col6α1^−/−^ ob/ob mouse model [40], Dr. Philipp E. Scherer’s laboratory showed that a collagen VI deficiency led to an uncontrolled adipocyte expansion and was paradoxically associated with substantial improvements in whole-body energy homeostasis under high-fat diet exposure or in the ob/ob model [40]. Scientists have postulated that an absence of the ECM constrains adipocyte expansion to favor lipid storage over ectopic lipid accumulation [40,41]. The results from collagen VI (col6α1^−/−^)-deficient mice demonstrated that obesity-associated collagen deposition promoted mammary tumor progression [42,43,44]. One limitation of these studies was that the mouse model was not adipocyte-specific; a reduced collagen VI expression in other cell types and tissues may contribute to these phenotypes in knockout mice.

The role of collagen VI in obesity-related ECM remodeling has been extensively studied [11,31]. As a fibrillar collagen, collagen VI plays a pivotal structural role in obesity-related fibrosis. Collagen VI is composed of N-terminal globular sub-domains, collagenous regions, and C-terminal globular sub-domains [45]. In obese adipose tissue, the microfibrils formed by collagen VI assemble a highly filamentous meshwork through an interaction with other ECM molecules to maintain the three-dimensional tissue architecture [11,31,42].

Collagen VI is an important driver of fibroblast activation during fibrosis. ECM remodeling induced by collagen IV is associated with the activation of the TGF-β pathway. During obesity development, TGF-β is released and activated to induce SMAD2/3 phosphorylation [46]. The activation of the TGF-β pathway induces adipocyte de-differentiation and further enhances ECM deposition [46,47]. In addition, collagen VI expression in adipose tissue correlates with macrophage infiltration; notably, both collagen VI levels and a macrophage accumulation are associated with the body mass index [48]. Chronic inflammation featuring increased macrophage infiltration and excessive cytokine expression further modulates the ECM turnover and contributes to the progression of fibrosis in adipose tissue [49,50,51].

Endotrophin (ETP), a non-collagenous fragment of collagen VI, is an emerging biomarker for obesity. Bone morphogenetic protein 1 metalloproteinase and MMP14 have been identified as the proteases to cleave the α3 chain of collagen VI and generate ETP fragments [31,33]. Elevated ETP levels in plasma correlate with an aberrant matrix structure in adipose tissue that occurs during excessive fat storage [31]. The overexpression of ETP in obese mice further augmented insulin resistance, characterized by a significant increase in lipolysis, inflammation, and cellular apoptosis in adipose tissue [11]. Macrophage marker F4/80 and the acute-phase inflammatory marker serum amyloid A3 were significantly increased in ETP-transgenic mice under a high-fat diet (HFD) condition; an elevated ECM deposition was also observed in these mice. These results suggest that ETP triggers obesity-associated inflammation and fibrosis in adipose tissue [48]. ETP binds to TEM8/ANTXR1 and Neuron-glial antigen 2 (NG2) to induce AKT and WNT signaling in cancer cells, subsequently promoting mammary tumor progression [42,45]. These results demonstrate that the cleavage of collagen VI in adipose tissue induces metabolic changes and promotes obese phenotypes by elevating ETP levels. Adipocyte-derived ETP promoted malignant tumor progression in a mouse model [52,53]. ETP levels are elevated in human breast cancer and colon cancer patients [52,53]. These results suggest that ETP partially mediates obesity-associated tumor progression. ETP can also be derived from liver tissue [54]; therefore, it may contribute to liver-associated cancer progression.

### 2.2. MMPs

MMPs cleave many non-ECM proteins and also degrade different components of the ECM, essential for tissue homeostasis [55]. An alteration to the MMP expression or in the balance between MMPs and their tissue-specific inhibitors is crucial for ECM remodeling during normal tissue development and disease progression. During obese development, an increased expression of MMP3, MMP11, MMP12, MMP13, and MMP14 has been detected and a reduction in MMP7, MMP9, MMP16, and MMP24 has been detected [55,56,57]. MMP13 cleaves interstitial collagens such as type I and III. MMP3 and MMP7 are mainly involved in fibronectin degradation [55,56,57]. Among these MMPs, MMP14 (MT1-MMP1) is the predominant pericellular collagenase in adipose tissue [55].

MMP14 is highly expressed in obese adipose tissue [33]. Local hypoxia in obese adipose tissue drives the activation of HIF-1α, which further induces MMP14 expression [33]. A variant in the human MMP14 is linked to obesity and diabetes [58,59]. The silencing of MMP14 leads to an impaired adipose tissue formation; it eventually led to severe lipodystrophy in mice [55,58]. One important function of MMP14 in adipose tissue is to digest collagen VI and generate ETP [33]. At the early stage of obesity development, MMP14 digests collagen and prevents fibrosis, which in turn promotes the healthy expansion of the tissue [33]. At the late stage of obesity, tremendous amounts of collagen accumulate in adipose tissue [33]. The local pathological changes ultimately lead to systemic insulin resistance and other metabolic disorders [33]. The role of MMP14 in cancer development and progression has been well-established [60]. However, how obesity-associated MMP14 expression in adipose tissue contributes to cancer progression remains to be determined.

### 2.3. Fibronectin and Others

Fibronectin was highly expressed in white adipose tissue derived from obese mouse models and patients [61]. The enhanced deposition of fibronectin has also been detected in mature 3T3-F442A adipocytes after co-culturing with breast cancer cells [62]. During obese development, fibronectin interacts with other ECM proteins such as collagen, which leads to an increased matrix stiffness and concomitantly altered biochemical signaling in adipocytes [61]. The function of fibronectin in tumor cell invasion is well-established [34,61,62]. It would be interesting to determine whether a fibronectin deposition in obese adipose tissue directly contributes to cancer progression.

Elastin is another structural ECM protein identified from adipose tissue [58]. Decreased elastin levels have been observed in obese WAT [58], probably due to MMP-dependent degradation [63]. Fibrillin-1 is one of the main components of microfibers, which have also been detected in adipose tissue [64]. Mutations in the fibrillin-1 gene lead to adipose tissue dysfunction and cause Marfan syndrome, marfanoid progeroid lipodystrophy syndrome, and neonatal progeroid syndrome [64]. However, the role of elastin and fibrillin-1 in obesity and obesity-associated cancer progression remains to be investigated.

## 3. Role of Fibroblasts and Adipocyte-Derived Myofibroblasts/Fibroblasts in Obesity-Associated Fibrosis

During the development of fibrosis, excess ECM proteins are produced whilst ECM degradation is limited. Several cell types in adipose tissue, including adipocyte progenitors, adipocytes, fibroblasts, and myofibroblasts, are responsible for the production of ECM proteins [65]. Fibroblasts and myofibroblasts are considered to be the dominant producers of the ECM in fibrosis-prone tissue [66,67]. A recent study showed an increased FSP1^+^ fibroblast population in adipose tissue during obesity [68]. FSP1^+^ fibroblasts promote adipogenesis by enhancing pre-adipocyte differentiation through the platelet-derived growth factor (PDGF) signal pathway, extracellular matrix remodeling, and YAP activation [68].

Recent studies have shown that adipocytes maintain plasticity and de-differentiate into pre-adipocytes and fibroblast-like cells [25,69]. The de-differentiation is characterized by a decreased lipid content and adipocyte size; a reduced expression of adipogenic differentiation markers such as PPAR-γ, perilipin and FABP4; and the enhanced expression of fibrotic genes/proteins such as collagen I, collagen VI, α-SMA, and FSP1. These changes in cell phenotypes and gene expression profiles indicate that de-differentiated adipocytes acquired myofibroblast/fibroblast phenotypes [70,71].

Adipocytes isolated from obese adipose tissue acquire a fibroblast-like transcriptional signature in response to a chronic high-fat diet [10]. These results suggest a potential role of obesity in promoting adipocyte de-differentiation. Multiple factors and microenvironmental cues may contribute to adipocyte de-differentiation, including PDGF, TGF-β, Wnts (Wnt1, 2 and 6), systemic metabolic stress, and hypoxia [10,20,25,72,73,74] (Figure 1). The PDGF pathway serves as a major regulator of fibroblast activation [20,72]. The binding of PDGF to PDGFRα activates multiple downstream signaling cascades, including MAPK and AKT/PI3K. The activated PDGFR pathway promotes the de-differentiation of adipocytes to fibroblasts/myofibroblasts [20,72]. The role of the TGF-β pathway in tissue fibrosis is well-established. An increased expression of TGF-β and the activation of its downstream pathway have been detected in obese adipose tissue [10,72]. The binding of TGF-β to the receptors TGFBR1 and TGFBR2 activates the canonical SMAD2/3 pathway, which subsequently induces the transcription of multiple ECM genes such as collagen I, collagen VI, and fibronectin [10,72]. Non-canonical TGF-β signaling involves the activation of RhoA and PI3K/AKT, which induces the contraction of collagen fibers during fibrosis. The aberrant activation of the Wnt pathway induces systemic insulin resistance and other metabolic disorders during obese development and also contributes to fibrosis in adipose tissue [73,74]. The binding of Wnt ligands to low-density lipoprotein receptors and Frizzled receptors induces the nucleus translocation of β-catenin. Transcription factor TCF/LEF forms complexes with β-catenin in the nucleus and activates the fibroblast gene signature [25,72].

Myofibroblasts/fibroblasts drive fibrosis by generating both biochemical components and biophysical cues. The connective tissue in adipose tissue is composed of an ECM deposited by myofibroblasts/fibroblasts, including collagen, fibronectin, elastin, and laminin [10,72]. The myofibroblasts/fibroblasts also “tug and pull” the ECM network around them, which further enhances the tissue-level mechanical forces and fibrosis in obesity [72]. The role of adipocyte-derived myofibroblasts/fibroblasts in obesity-associated fibrosis has recently been appreciated [7]. These cells produce significant amounts of ECM proteins and secret many cytokines and growth factors [24]. These cytokines and ECM molecules create microenvironmental niches that recruit a variety of stromal cells, including, but not limited to, macrophages, eosinophils, and innate lymphoid 2, subsequently generating chronic inflammation in adipose tissue [24].

## 4. Obesity-Associated Fibrosis and ECM Remodeling in Cancer Progression

Obesity is a well-established risk factor for many types of cancer. Obesity induces both local and systemic changes that contribute to multiple stages of cancer development and progression, which has been demonstrated in multiple mouse models [15,40,75]. Circulating insulin and insulin-like growth factor-1 have been identified as factors that mediate obesity-dependent cancer progression [75,76]. At a tissue level, chronic inflammation and hypoxia-induced angiogenesis in obese adipose tissue also contribute to the tumor progression [77]. However, the role of obesity-associated fibrosis in cancer development and progression has not been appreciated until recently. In addition, breast cancer and several other cancers such as epithelial ovarian cancer [78] and gastrointestinal cancer [79] form in proximity to adipocytes. One report showed that obesity-dependent ECM deposition in liver tissue contributed to hepatocarcinoma development [80]. However, most previous studies have focused on the role of ECM remodeling in adipose tissue in obesity-associated cancer progression [15,40,81,82].

Obesity-associated fibrosis in adipose tissue, featuring enhanced ECM deposition and crosslinking, promotes cancer development and progression at different stages [14,15] (Figure 2). A recent study showed that an ECM isolated from obese adipose tissue stimulated tumor cell growth and drove the transformation of premalignant human epithelial cells [15]. Adipose-derived myofibroblasts deposit fibronectin, type I collagen, and type VI collagen with a more linearized structure, subsequently enhancing ECM stiffness [15]. These biochemical and biophysical changes in the ECM further promote the malignant transformation and enhance the tumor growth [15]. A recent study showed that obesity-associated ECM remodeling enhanced cancer cell migration, invasion, and metastasis [14]. Increased collagen VI in adipose tissue enhances ECM deposition and alignment, subsequently inducing the invasiveness of mammary tumors [14]. Obesity-associated ECM remodeling has also been detected in human breast cancer. Tumors from obese women exhibited an enhanced expression of fibronectin and smooth muscle actin. An increased thickness and alignment of collagen fibers were also detected in breast cancer from obese women compared with cancer from lean women [15]. Stromal cells in adipose tissue play important roles in obesity-associated ECM remodeling and cancer progression. A recent study showed that stromal cells derived from obese adipose tissue facilitated tumor cell migration by enhancing the contraction of local collagen fibers [83].

Biochemical and biophysical cues derived from the ECM in adipose tissue modulate multiple pathways in cancer cells. An increased matrix stiffness in obese mammary WAT directly activates transcription factor YAP/TAZ, which in turn induces tumor cell growth [15]. The aberrant activation of YAP/TAZ enhances tumor-initiating cell activity and promotes tumor cell growth [15]. NG2/EGFR and MAPK are other pathways that are regulated by obesity-associated ECM remodeling. Collagen VI in adipose tissue induces the proteoglycan NG2/EGFR and β1 integrin pathways, subsequently driving the MAPK signaling downstream, leading to cancer cell adhesion, migration, and metastasis [14].

Fibrosis in obese adipose tissue is associated with the activation of myofibroblasts/fibroblasts and the infiltration of macrophages. These cellular components also promote cancer development and progression by secreting a variety of growth factors and cytokines to induce angiogenesis and immune evasion [49,50,51]. The function and regulation of myofibroblasts/fibroblasts and macrophages in the tumor microenvironment has been extensively studied and well-summarized [84,85,86].

## 5. Conclusions and Future Perspectives

Obesity induces profound changes in adipose tissue, including metabolic reprogramming, fibrosis, and chronic inflammation. The current line of evidence suggests that adipocyte de-differentiation and myofibroblasts/fibroblasts play crucial roles in ECM remodeling in response to obesity. It is very likely that the de-differentiation process is orchestrated by a cascade of complex modulators. However, the core regulatory network that triggers adipocyte dedifferentiation has not been fully characterized. Several questions remain to be addressed, including whether adipocytes have an equal capability to undertake de-differentiation or whether they need to be in an already altered state; whether and how tumor cells regulate adipocyte de-differentiation and fibrosis in adipose tissue; and how the direct and indirect cancer cell–adipocyte interaction differentially modulates cancer progression. Addressing these questions may identify the process of adipocyte de-differentiation and obesity-associated ECM remodeling as novel targets for anti-fibrotic and anti-cancer therapies.

## Figures and Tables

**Figure 1 cancers-14-05684-f001:**
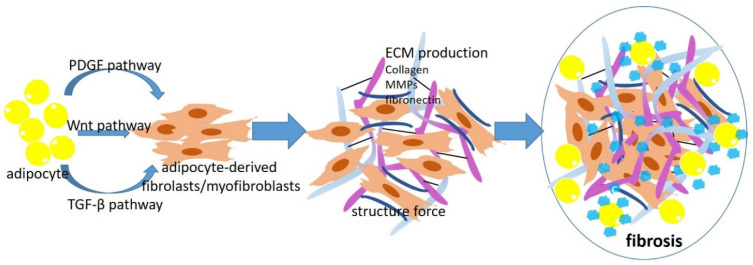
The mechanism underlying adipocyte-derived myofibroblasts/fibroblasts in obesity-associated fibrosis. During obesity development, adipocytes acquire a fibroblast-like transcriptional signature through PDGF, WNT, and TGF-β pathways. Adipocyte-derived fibroblasts/myofibroblasts deposit significant amounts of ECM and ECM-related enzymes, including collagen, fibronectin, and MMPs. The fibroblasts/myofibroblasts also “tug and pull” the ECM network around them, which further enhances the ECM alignment and fibrosis.

**Figure 2 cancers-14-05684-f002:**
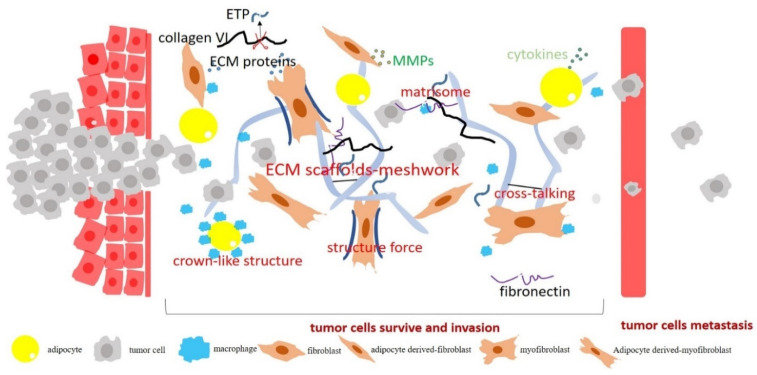
The role of obesity-associated fibrosis and ECM remodeling in cancer progression. During obesity development, adipocytes transform or de-differentiate into myofibroblast/fibroblast-like cells. These cells produce significant amounts of collagen VI, fibronectin, and MMPs. Obesity-associated fibrosis stimulates tumor cell growth, drives the transformation of premalignant cells, and induces tumor invasion and metastasis. Obesity-associated accumulation of myofibroblasts/fibroblasts and immune cells in adipose tissue releases cytokines and pro-inflammation factors, which further enhances cancer progression.

**Table 1 cancers-14-05684-t001:** The main components in obesity-related ECM remodeling.

Component	In Obesity	In Cancer	Function (In Obese)	Ref.
Collagen VI	Increased	Increased	Maintains 3D architecture; cellular function	[14,40,46,47]
MMP14	Increased	Increased	Digests collagen; affects pre-adipocyte differentiation	[33]
Fibronectin	Increased	Increased	Structure support	[34,61,62]
Elastin	Decreased	Increased	Structure support	[58]
Fibrillin-1	Increased	Increased	Forms microfibrils	[64]

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
