# Peer review of "Obesity-Associated ECM Remodeling in Cancer Progression"

_cancers, 2022, doi:10.3390/cancers14225684_

Round 1
Reviewer 1 Report
The review addresses an important topic regarding the role of extracellular matrix in carcinogenesis from the perspective of obesity as a potential driver. A number of relevant ECM molecules are discussed that are altered in obese adipose tissues, and independently many of these have known roles in cancer, but the weak link is in connecting these topics. Fibrosis is another major focus, which is clearly relevant to obesity and carcinogenesis, but apart from presenting some clear links in the context of breast cancer, the review often fails to build a strong case that ECM is the major mechanism linking obesity-dependent fibrosis with carcinogenesis or tumor progression. Several issues need to be addressed:
1. Several mechanisms have been defined by which obesity stimulates carcinogenesis and cancer progression. Modulation of ECM is one such mechanism, but the authors should begin their review by giving readers a context to understand how their review topic differs from other major obesity-related carcinogenesis mechanisms such as obesity-induced inflammation, pro-cancer metabolites, microbiome changes, and cancer promoting growth factors such as IGF1. One major distinction is that the latter mechanisms are generally systemic, whereas ECM effects are local. This topic is presented beginning on line 229 as it relates to fibrosis. To give proper context for the present review, these systemic mechanisms should be summarized in the introduction, and then discuss which systemic factors act in part through ECM modulation where relevant to specific mechanisms discussed in the review.
2. Relevant to the previous comment, the authors discuss some specific roles of adipocytes that are in the nascent tumor microenvironment, but it is confusing when they discuss adipose tissue in general, which may be distant from the tumor. Some discussion of which cancers (eg breast) form in the proximity of adipose tissue would be useful. Which other cancers form in proximity to adipocytes? Do obesity-dependent changes in ECM play any known role in carcinogenesis that occurs in tissues that lack adipocytes?
3. The Collagen section discusses roles of collagen VI in obesity-induced fibrosis. However, apart from briefly mentioning its link to chronic inflammation, it is unclear how this section relates to the topic of carcinogenesis. The relevance of the fragment ETP to cancer seems weak. Does ETP act systemically or only if a malignant cell originates in adipose tissue?
4. The discussion of MMPs and their inhibitors in adipose tissue needs work to explain why it is relevant to cancer. Clearly, MMPs are important in cancer, but the only specific example cited is for MMP14 expressed by prostate cancer cells, not adipocytes. This does not justify the concluding statement that adipose MMP14 somehow contributes to tumor invasion.
5. In general, the concluding paragraph lacks a clear presentation of major challenges for advancing the field. The authors conclude “it is not clear whether inflammation is the driver or rather a consequence of ECM remodeling.” Indeed, there is strong evidence for causality in both directions that should be cited. This topic should be expanded to become a major section of the review.
6. The text needs editing for a number of grammar and usage errors. For example, line 83 should read “Collagens are major structural ECM components in adipose tissue. Increased expression,,, have been detected…” Editing by a native English speaker is needed.
Author Response
Response to Reviewer 1
- Several mechanisms have been defined by which obesity stimulates carcinogenesis and cancer progression. Modulation of ECM is one such mechanism, but the authors should begin their review by giving readers a context to understand how their review topic differs from other major obesity-related carcinogenesis mechanisms such as obesity-induced inflammation, pro-cancer metabolites, microbiome changes, and cancer promoting growth factors such as IGF1. One major distinction is that the latter mechanisms are generally systemic, whereas ECM effects are local. This topic is presented beginning on line 229 as it relates to fibrosis. To give proper context for the present review, these systemic mechanisms should be summarized in the introduction, and then discuss which systemic factors act in part through ECM modulation where relevant to specific mechanisms discussed in the review.
Response: This point is well taken. We now include the following sentence in the first paragraph:
“The obesity-dependent cancer development and progression involve inflammation, adipokines, microbiota, hyperinsulinaemia and IGF-1 signaling, which has been extensively summarized and reviewed (3-5). In this review, we focused on recent progress in the function and regulation of obesity-associated ECM remodeling and adipocyte plasticity during cancer progression.”
In part 4, we also include several sentences to summarize the established mechanism of obesity-dependent cancer progression, “Circulating insulin and insulin-like growth factor-1 have been identified as obesity-induced factors that promotes cancer progression (78,79). At the tissue level, chronic inflammation and hypoxia-induced angiogenesis in obese adipose tissue also contributes to tumor progression (80). However, roles of obesity-associated fibrosis in cancer development and progression have not been appreciated until recently.”
- Relevant to the previous comment, the authors discuss some specific roles of adipocytes that are in the nascent tumor microenvironment, but it is confusing when they discuss adipose tissue in general, which may be distant from the tumor. Some discussion of which cancers (eg breast) form in the proximity of adipose tissue would be useful. Which other cancers form in proximity to adipocytes?
Response: We now include the information about cancer types (breast, ovarian, and colon) in the proximity of adipose tissue in first paragraph of Part 4. Because majority of previous studies used mammary tumor as a model to study obesity-associated ECM remodeling, this review focused on the local effect of adipose tissue on breast cancer progression.
Do obesity-dependent changes in ECM play any known role in carcinogenesis that occurs in tissues that lack adipocytes?
Response: We only identified one report showing that obesity-associated ECM deposition contributes to liver cancer progression (83). Most of previous studies are about function and regulation ECM remodeling in adipose tissue. This comment suggests a potential future direction for obesity-associated cancer progression research.
- The Collagen section discusses roles of collagen VI in obesity-induced fibrosis. However, apart from briefly mentioning its link to chronic inflammation, it is unclear how this section relates to the topic of carcinogenesis.
Response: Collagen VI is highly expressed in obese adipose tissue. Dr. Philipp E. Scherer’s laboratory showed that knockout of collagen VI inhibited obesity-associated mammary tumor development and progression (43-45). These results suggest that increased collagen VI deposition in adipose tissue is crucial for obesity-associated mammary tumor progression.
The relevance of the fragment ETP to cancer seems weak. Does ETP act systemically or only if a malignant cell originates in adipose tissue?
Response: ETP is a fragment of collagen VI. It has been shown that adipocyte-derived ETP promotes malignant tumor progression (53, 54). ETP serum levels were elevated in in human breast cancer and colon cancer patients (54), suggesting that human ETP also contribute to tumor progression (53,54). ETP can also be produced from other tissues, such as liver tissue (54). This information is now included in second and fifth paragraph from Part 2.
- The discussion of MMPs and their inhibitors in adipose tissue needs work to explain why it is relevant to cancer. Clearly, MMPs are important in cancer, but the only specific example cited is for MMP14 expressed by prostate cancer cells, not adipocytes. This does not justify the concluding statement that adipose MMP14 somehow contributes to tumor invasion.
Response: We mainly focused on MMP-dependent ECM remodeling in adipose tissue under obese condition. MMP14 is dramatically upregulated in obese adipose tissue. It was reported that MMP14 cleaves collagen VI and generates ETP (34), which is a driver of breast cancer (53, 54).
Roles of MMP14 in cancer development and progression have been well-established (61). However, how obesity-associated MMP14 expression in adipose tissue contributes to cancer progression remains to be determined.
We agree with the reviewer that MMP14 expression by cancer cell cannot justify the function of adipose MMP14; this sentence and reference has been removed.
We have revised and included additional information in the sixth paragraph of Part 2.
- In general, the concluding paragraph lacks a clear presentation of major challenges for advancing the field. The authors conclude “it is not clear whether inflammation is the driver or rather a consequence of ECM remodeling.” Indeed, there is strong evidence for causality in both directions that should be cited. This topic should be expanded to become a major section of the review.
Response: Thanks for bring this to our attention. We now delete the sentence ‘it is not clear whether inflammation is the driver or rather a consequence of ECM remodeling’. We agree with the review that this topic is important and could be expanded to become a major section. In the current manuscript, we mainly focused on recent progress in the obesity-related ECM remodeling and adipocyte plasticity. We suggest discussing and summarizing this topic in the future review article.
- The text needs editing for a number of grammar and usage errors. For example, line 83 should read “Collagens are major structural ECM components in adipose tissue. Increased expression,,, have been detected…” Editing by a native English speaker is needed.
Response: Thanks for bring this to our attention. The manuscript has been edited by a native English speaker.
Reviewer 2 Report
In this manuscript, the authors propose a review on recent published data related to the impact of obesity on extracellular matrix remodeling, especially the adipocyte dedifferentiation into fibroblast-like phenotype, and the consequences on extracellular matrix proteins such as collagens (deposition and organization) and cancer progression. The review is very interesting and is mostly based on extensive work on the latest findings in the field. However, the manuscript cannot be accepted for publication in its current form and some issues should be addressed. Of course, some of my comments can also be discussed.
1) In the third paragraph of the introduction section (line 48), as well as in the other sections of the manuscript, we clearly understand the message underlining the impact of obesity on the remodeling of the extracellular matrix via the differentiation of adipocytes into fibroblast-like phenotype, which in turn will impact extracellular matrix. However, it seems that a link is missing, especially the mechanisms related to the dedifferentiation process of adipocytes. Do the tumor cells induce this dedifferentiation in a cross-talk process? this of course seems to be the case. Therefore, a more detailed part on the role of tumor cells in the process of adipocyte dedifferentiation could significantly improve the impact of the review. The authors have mentioned briefly this point in reference 19.
2) It should be interesting to discuss briefly, whether direct mechanisms in the cross-talk between stromal cells (adipocytes) and tumor cells could contribute to tumor progression (Ling et al., 2020).
Ling L, Mulligan JA, Ouyang Y, Shimpi AA, Williams RM, Beeghly GF, Hopkins BD, Spector JA, Adie SG, Fischbach C. Obesity-associated Adipose Stromal Cells Promote Breast Cancer Invasion Through Direct Cell Contact and ECM Remodeling. Adv Funct Mater. 2020; 30(48):1910650. doi: 10.1002/adfm.201910650.
3) In the same way, the effect of extracellular matrix components such as type I collagen on the adipose stromal cells phenotype and differentiation could be mentioned (Ikejima et al., 2020 ; Seo et al., 2020).
Liu X, Long X, Gao Y, Liu W, Hayashi T, Mizuno K, Hattori S, Fujisaki H, Ogura T, Onodera S, Wang DO, Ikejima T. Type I collagen inhibits adipogenic differentiation via YAP activation in vitro. J Cell Physiol. 2020; 235(2):1821-1837. doi: 10.1002/jcp.29100.
Seo BR, Chen X, Ling L, Song YH, Shimpi AA, Choi S, Gonzalez J, Sapudom J, Wang K, Andresen Eguiluz RC, Gourdon D, Shenoy VB, Fischbach C. Collagen microarchitecture mechanically controls myofibroblast differentiation. Proc Natl Acad Sci USA. 2020; 117(21):11387-11398. doi: 10.1073/pnas.1919394117.
4) In the section on collagens, type I, III and VI collagens seem to be overexpressed in the microenvironment of tumors associated with obesity. Unless I am mistaken, references 11 and 12 are not related to type III collagen. However, a recent study has reported that type III collagen play a crucial role in cell dormancy and consequently long-term metastasis. In this study, the authors underlined the importance of type III collagen organization in such process. However, the data showed that this collagen was produced by cancer cells, and not the stromal ones (Di Martino et al., 2022).
Di Martino JS, Nobre AR, Mondal C, Taha I, Farias EF, Fertig EJ, Naba A, Aguirre-Ghiso JA, Bravo-Cordero JJ. A tumor-derived type III collagen-rich ECM niche regulates tumor cell dormancy. Nat Cancer. 2022 ; 3(1):90-107. doi: 10.1038/s43018-021-00291-9.
Author Response
Thank reviewers for their insightful comments and suggestions on our manuscript.
1)In the third paragraph of the introduction section (line 48), as well as in the other sections of the manuscript, we clearly understand the message underlining the impact of obesity on the remodeling of the extracellular matrix via the differentiation of adipocytes into fibroblast-like phenotype, which in turn will impact extracellular matrix. However, it seems that a link is missing, especially the mechanisms related to the dedifferentiation process of adipocytes. Do the tumor cells induce this dedifferentiation in a cross-talk process? this of course seems to be the case. Therefore, a more detailed part on the role of tumor cells in the process of adipocyte dedifferentiation could significantly improve the impact of the review. The authors have mentioned briefly this point in reference 19.
Response: This point is well taken. The dedifferentiation is largely induced by the cancer cell-adipocyte interaction (21, 22). The mechanisms of the dedifferentiation process of adipocytes were detailed summarized in part 3.
2) It should be interesting to discuss briefly, whether direct mechanisms in the cross-talk between stromal cells (adipocytes) and tumor cells could contribute to tumor progression (Ling et al., 2020).
Ling L, Mulligan JA, Ouyang Y, Shimpi AA, Williams RM, Beeghly GF, Hopkins BD, Spector JA, Adie SG, Fischbach C. Obesity-associated Adipose Stromal Cells Promote Breast Cancer Invasion Through Direct Cell Contact and ECM Remodeling. Adv Funct Mater. 2020; 30(48):1910650. doi: 10.1002/adfm.201910650.
Response: This point is well taken. We have include this discussion marked with yellow in second paragraph of Part 4.
3) In the same way, the effect of extracellular matrix components such as type I collagen on the adipose stromal cells phenotype and differentiation could be mentioned (Ikejima et al., 2020 ; Seo et al., 2020).
Liu X, Long X, Gao Y, Liu W, Hayashi T, Mizuno K, Hattori S, Fujisaki H, Ogura T, Onodera S, Wang DO, Ikejima T. Type I collagen inhibits adipogenic differentiation via YAP activation in vitro. J Cell Physiol. 2020; 235(2):1821-1837. doi: 10.1002/jcp.29100.
Seo BR, Chen X, Ling L, Song YH, Shimpi AA, Choi S, Gonzalez J, Sapudom J, Wang K, Andresen Eguiluz RC, Gourdon D, Shenoy VB, Fischbach C. Collagen microarchitecture mechanically controls myofibroblast differentiation. Proc Natl Acad Sci USA. 2020; 117(21):11387-11398. doi: 10.1073/pnas.1919394117.
Response: Thank you for your suggestion. We have added this discussion marked with yellow in second paragraph of Part 2.
4) In the section on collagens, type I, III and VI collagens seem to be overexpressed in the microenvironment of tumors associated with obesity. Unless I am mistaken, references 11 and 12 are not related to type III collagen. However, a recent study has reported that type III collagen play a crucial role in cell dormancy and consequently long-term metastasis. In this study, the authors underlined the importance of type III collagen organization in such process. However, the data showed that this collagen was produced by cancer cells, and not the stromal ones (Di Martino et al., 2022).
Di Martino JS, Nobre AR, Mondal C, Taha I, Farias EF, Fertig EJ, Naba A, Aguirre-Ghiso JA, Bravo-Cordero JJ. A tumor-derived type III collagen-rich ECM niche regulates tumor cell dormancy. Nat Cancer. 2022 ; 3(1):90-107. doi: 10.1038/s43018-021-00291-9.
Response: We are sorry for this oversight. This mistake has been corrected. We are aware of the research article recently published on Nature Cancer. However, this study is not related to obesity and adipocytes, and we did not include it in this review.
Round 2
Reviewer 1 Report
Most of the previous conceptual/technical concerns were addressed. Although the response states that the manuscript was edited by a native English speaker, numerous single/plural errors and incorrect use of articles throughout indicates that the editor is not proficient in written English. The numerous errors will distract readers from the important concepts that authors intend to convey. The following is an incomplete list of usage and syntax errors.
Line 95: replace ‘while promotes’ with ‘and promotes’ or ‘while promoting’
Line 98: replace ‘mice’ with ‘mouse’
Genotypes should be italicized throughout
Line 115: ‘Collagen VI is the driver of fibroblast activation during fibrosis.’ This overstates the role of collagen VI by implying it is the only driver. Clearly, multiple secreted factors drive fibroblast activation, and even in the context of ECM, other drivers including tenascin C play important roles (eg PMID: 35400564).
Lines 146-147: ‘MMPs degrade different components of ECM and also cleave many non-ECM proteins, which are essential for tissues homeostasis (56).’ As written, the authors are stating that non-ECM proteins are essential for tissue homeostasis. If the intent is to state that MMP cleavage is essential, replace ‘, which’ with ‘that’
Line 147 ‘alternation’ should be ‘alteration’
Line 157: replace ‘A genetic variant in the human MMP14’ with ‘A variant in the human MMP14 gene’
Line 158: replace ‘Silence’ with ‘Silencing’
Line 171: replace ‘adipocyte’ with ‘adipocytes’
Line 180: ‘was’ ‘were’
Line 183 ‘remains’ ‘remain’
Line 214: ‘Binding of TGFβ to the receptor’ Which receptor? TGFβ has several receptors with divergent functions.
Line 230 ‘amount’ ‘amounts’
Lines 245-246: ‘In addition breast cancer, there are some cancers forming in proximity to adipocytes’ replace with ‘In addition to breast cancer, several other cancers form in proximity to adipocytes’
Lines 294-295z: ‘There are several questions remain to be addressed’ should read “Several questions remain to be addressed’
Line 316: sentence fragment ‘The authors no conflicts of interest.’
Throughout this review, the authors frequently use passive voice sentence construction (e.g. “It has been shown that…”). Active voice should be used in most cases. See: https://writing.wisc.edu/handbook/style/ccs_activevoice/
Author Response
We are really grateful to the review for his/her time and effort to edit the English. We have revised the manuscript according to the review’s comments.
Reviewer 2 Report
The authors have addressed all the issues I have raised in an appropriate way.
Author Response
NA
Round 3
Reviewer 1 Report
The remaining grammar and usage issues were addressed.